# Design and Verification of Humidity Sensors Based on Magnesium Oxide Micro-Arc Oxidation Film Layers

**DOI:** 10.3390/s20061736

**Published:** 2020-03-20

**Authors:** Mingqiang Pan, Jun Sheng, Jizhu Liu, Zeming Shi, Lei Jiu

**Affiliations:** 1School of Mechanical and Electric Engineering, Soochow University, Suzhou 215123, China; pmqwl@126.com (M.P.); 20195229044@stu.suda.edu.cn (Z.S.); 20175229049@stu.suda.edu.cn (L.J.); 2Jiangsu Provincial Key Labor Atory of Advanced Robotics, Soochow University, Suzhou 215123, China; 3Robotics and Microsystems Center, Soochow University, Suzhou 215123, China

**Keywords:** humidity range, MgO, micro-arc oxidation, response signal

## Abstract

Humidity detection range is an important indicator for measuring the performance of humidity sensors, but semiconductor humidity sensors often face the problems of narrow detection ranges and insufficient detection sensitivities. In this paper, a magnesium oxide (MgO) humidity sensor based on micro-arc oxidation (MAO) technology was designed to solve these problems by simultaneously using impedance and capacitance as the response signals, as well as by normalizing the output of the two signals. The experimental results showed that the average output of the micro-arc MgO ceramic film, with impedance as the response signal, could reach 150 in the low relative humidity(RH) range (11.3–67% RH), which was much higher than its sensitivity in the high humidity range (< 1), and the film showed fast response (13 s) and recovery (61 s). Under high humidity conditions (67–97.3% RH), with capacitance as the response signal, the output of the micro-arc MgO was as high as 120. Therefore, the micro-arc MgO humidity sensor with impedance, and the sensor with capacitance as the response signal, demonstrated good stability in low humidity and in high humidity environments, respectively, indicating that the method of selecting appropriate response signals for different humidity environments can be applied to extend the humidity detection range of sensing material, and to improve the humidity detection capability of a sensor.

## 1. Introduction

Humidity sensors have been widely used in various fields of human production and life, including weather forecasting, health care, and food processing [1,2,3,4,5,6]. Relative humidity (RH) detection range is an important indicator to measure the detection performance of humidity sensors. It is often utilized differently in different fields. Many humidity sensors currently exhibit distinct moisture sensitivity responses in different humidity environments. Yin et al. [7] designed a 3D pillared-layer cadmium (II) metal organic framework, the impedance change of which was greater than two orders of magnitude in the range of 54–97% RH. However, no significant change in impedance was found in the low humidity range, indicating it was only suitable for medium and high humidity environment testing. Juhász et al. [8] used an anodic oxidation technique to prepare a porous alumina humidity sensor with sensitivity up to 27 pF/RH% in the high humidity range, and merely 2.5 pF/RH% in the low humidity range. The development of a humidity sensor with a wide range of detection capability has become a research hotspot. At present, most researchers focus on the modification of materials and microstructure [9,10,11,12]. Few paid attention to the influence of the choice of response signals on the performance of humidity sensors [13,14,15].

Micro-arc oxidation (MAO) is a new technology to directly grow ceramic layers in situ on the surface of non-ferrous metals. This is done by placing a valve metal (such as Al, Mg, Ti, or their alloys) in an aqueous electrolyte solution, and using electrochemical methods to generate spark discharge points in the surface micropores of the material, which is a method for generating oxides under the influence of thermochemistry, plasma chemistry, and electrochemistry. Technologies like this have the advantages of easy operation, environmental protection, and high efficiency [16,17]. Meanwhile, MgO as a transition metal oxide is rarely used as a sensing material, and the main reason is that it has a larger band gap, which makes it more difficult to generate carriers. However, its faster carrier mobility gives it more potential to develop a humidity sensor with a rapid response [18,19].

To this end, this paper is devoted to the use of MAO to grow MgO film layers in situ on the surface of Mg plates to produce humidity sensing material. The microstructure and composition of the oxide layer were characterized by scanning electron microscopy (SEM) and energy dispersive X-ray spectroscopy (EDS). The phase composition of the oxide layer was analyzed by X-ray diffraction (XRD). At the same time, impedance and capacitance were selected as the response signals to detect the humidity sensing performance of the film layer. The experimental results showed that the micro-arc MgO film with a porous structure on the surface had a large specific surface area. In different humidity environments, the micro-arc MgO film layer with impedance and capacitance as the response signals showed a major difference in output, indicating that the method of selecting appropriate response signals for different humidity conditions could be used to extend the humidity detection range of sensing material.

## 2. Materials and Methods

### 2.1. Materials Synthesis

The specific manufacturing steps (as shown in Figure 1) of the sensor were as following:
(a)First, a 20 × 20 × 5 mm magnesium plate sample was cut out (purity > 99.5%). Sandpaper (400 # → 600 # → 800 #) was used to grind and remove oil (10% NaOH solution), then the magnesium plate was rinsed with deionized water and air-dried for later use.(b)A 10 g/L Na_2_SiO_3_ solution using deionized water was prepared, and then NaOH powder was added to adjust the pH of the electrolyte to 13.(c)The sample was hung on the electrode and immersed into the electrolyte. The anode was a magnesium plate and the cathode was a stainless steel electrolytic cell.(d)The power supply parameters were adjusted. The average current density was 4A/cm2, the duty cycle was 35%, the frequency was 550 Hz, the temperature of the electrolyte was maintained at 10–40 °C, and the oxidation time was 20 min. A circulating cooling device was used to keep the temperature of the configured electrolyte at 10–40 °C.(e)After oxidation, the micro-arc MgO plate was rinsed thoroughly with deionized water and dried in hot air.(f)A 200 # interdigital electrode screen plate was designed and purchased. The test piece was placed under the mesh, and the prepared conductive silver paste was uniformly printed on the micro-arc oxidation plate through the screen hole by using a scraper.(g)After the printing was completed, the MAO plate was held at 80 °C for 1 h to ensure that the silver paste was completely cured to fabricate a micro-arc MgO humidity sensor.


### 2.2. Fabrication of Humidity Sensor

As shown in Figure 2, the glass chamber was double-layered, and the inter layer was filled with plastic foam as a moisture barrier. The humidity in the glass chamber was controlled by the evaporation of water. The glass chamber had a built-in thermo-hygrograph as an indicator of ambient temperature and humidity. The entire unit could stably obtain a humidity range of 20–80% RH. The signal electrode and signal receiving device (Agilent 33210A, 34410A) (Agilent, Suzhou, China) of the fabricated micro-arc MgO humidity sensor were connected with a computer. An AC voltage of 5 V was applied to obtain the impedance characteristics thereof.

Capacitance testing was performed using an LCR bridge tester (VC4090A) (Victor, Suzhou, China) at an operating frequency of 1 kHz. The entire device could continuously change the humidity in the test environment, which was mainly used to test the change in the two response signals (impedance and capacitance) of the sensor with humidity.

In addition, this paper configured LiCl (11.3% RH), CH_3_COOK (22.5% RH), MgCl_2_ (32.8% RH), K_2_CO_3_ (43.2% RH), NaBr (57.6% RH), NaCl (75.3% RH), KCl (84.2% RH), and K_2_SO_4_ (97.3% RH) of 25 °C saturated solution in order to obtain a stable humidity environment [20]. This particular solution was mainly used to test the response time and recovery time, wet hysteresis, and stability of sensors.

## 3. Results and Discussion

### 3.1. Characterization Results

The surface morphology and microstructure of the micro-arc MgO film layer were characterized by SEM, as shown in Figure 3a–c.

The surface and cross section of the micro-arc MgO film layer were shown at different magnifications. The thickness of the micro-arc oxide film was about 32 μm, which had good continuity, while the surface had a large number of abnormal pores with a diameter of 4–8 μm. This microporous structure allowed the micro-arc MgO film layer to have a larger surface area, providing more adsorption sites and reaction channels for water molecules, which was helpful for enhancing the moisture-sensing properties of the material. Figure 3d is an EDS diagram of the film layer, showing that the main components are Mg and O elements.

The XRD measurement of the micro-arc MgO film layer was performed with an X-ray diffractometer. The XRD results (Figure 4) showed that the MAO coating was mainly composed of MgO and Mg, where MgO was the reaction product in the MAO process and the main phase in the MAO coating.

### 3.2. Humidity-Sensing Properties

Figure 5a shows the relationship between different frequency conditions with the impedance of micro-arc MgO as a function of humidity, and it can be seen that at each frequency, the impedance of the micro-arc MgO decreases with the increase of humidity. However, as the frequency increased, the magnitude of this change decreased more significantly. Considering that micro-arc MgO with low frequency tests had higher sensitivity, excessively low frequency was expected to increase the complexity of signal conditioning. At the same time, excessive impedance at low frequencies required complex amplification circuits and rigorous application test conditions, resulting in low practicality. Therefore, the operating frequency of 1 kHz was selected to measure the humidity-sensing performance of the micro-arc MgO. Figure 5b shows that the micro-arc MgO humidity sensor exhibited good stability at the operating frequency of 1 kHz. Especially under high humidity conditions, there were no significant fluctuations in the impedance measured for multiple times.

At the 1 kHz operating frequency, the impedance and capacitance of the film in different humidity environments were measured separately to investigate the effects of different response signals on the humidity sensing performance of micro-arc MgO. As shown in Figure 6a, when RH rose from 11.3% to 97.3%, the impedance was used as the detection signal. In the low humidity environment, the impedance of the micro-arc MgO decreased exponentially with the increase in humidity. However, as the humidity continued to increase, the rate of decrease in the impedance of the film was significantly reduced, and finally there was no significant change. When the capacitor was used as the response signal, the capacitance rose slowly in the low humidity environment. As the humidity continued to increase, the film capacitance rose significantly and eventually increased exponentially. In order to find out the best switching point between the two signals, the signal outputs of the sensor were standardized and expressed by S_R_ and S_C_:
(1)$$S_{R}=\frac{R_{\mathrm{RH}}-R_{97.3\%\mathrm{RH}}}{R_{97.3\%\mathrm{RH}}}$$
(2)$$S_{C}=\frac{C_{\mathrm{RH}}-C_{11.3\%\mathrm{RH}}}{C_{11.3\%\mathrm{RH}}}$$

S_R_ and S_C_ denote the output result of micro-arc MgO when impedance and capacitance are used as response signals. R_RH_ and C_RH_ are the impedance and capacitance of micro-arc MgO in various humidity environments, respectively. R_97.3%_ and C_11.3%_ represent the impedance and capacitance values of the micro-arc MgO at 97.3% RH and 11.3% RH, respectively.

Figure 6b shows the output curve of micro-arc MgO with two kinds of response signals used. The intersection point of the curves is point A, and the corresponding humidity is about 67%. It can be seen that at relative humidity of 11.3–67%, when the response signal of the micro-arc MgO humidity sensor was impedance, the average output result could reach 150. On the contrary, at a relative humidity of 67–97.3%, the average output of a micro-arc MgO humidity sensor with capacitance as the response signal was about 120, which was much higher than the output of micro-arc MgO using impedance as the response signal (< 1). Obviously, when the impedance was used as the response signal, the micro-arc MgO had a high response in a low humidity environment. The micro-arc MgO humidity sensor with capacitance as the detection signal was more suitable for application under high humidity conditions. The relative humidity in the two response signals’ output change led to turning points at 67%. Designing a condition circuit to switch the response signal of the MgO film is key to the practical application of the above ideas. At present, there are related studies on this part. For example, Xin et al. [21] designed a signal switching circuit to switch signals through multi-channel gated switches. Realized real-time display and switching of measurement signals and observation signals. The above shows that it is feasible to improve the detection performance of the humidity sensor by means of signal switching.

Figure 7a,b demonstrates the response and recovery characteristics of micro-arc MgO. Relevant parameters were measured in a humidity bottle filled with a saturated solution. The environmental change from 11% RH to 75% RH was mainly achieved by rapidly transferring the micro-arc MgO plate along the route of LiCl (11.3% RH) → NaCl (75.3% RH) → LiCl (11.3% RH), thereby achieving the measurement of response time and recovery time. The response time and recovery time of the micro-arc MgO humidity sensor with impedance and capacitance as the response signal were 13 and 67, and 341 and 61 s, respectively. At the same time, this paper compared the response time and recovery time of other reported humidity sensors (see Table 1), which could prove that when the impedance was used as the response signal, the micro-arc MgO humidity sensor had faster response and recovery. However, the response time and recovery time of some excellent commercial humidity sensors could be controlled within 10s. Therefore, the response time and recovery time of the micro-arc MgO humidity sensor need to be further optimized.

Figure 7c,d show the hysteresis of the two sensors when the humidity rises from 11.3% RH to 97.3% RH and then drops to 11.3% RH. The micro-arc MgO humidity sensor with impedance as the response signal has a hysteresis of about 3.8% at 47% RH. When the capacitance is used as the response signal, the hysteresis calculated by the sensor at 78% RH is about 5.7%. Further, we compared the hysteresis of the designed humidity sensor with other types of humidity sensors (as shown in Table 2). It can be seen that the micro-arc MgO humidity sensor has low hysteresis.

Figure 8a,b represent the exponential fitting of the response curves of the two humidity sensors, respectively. The response signals of the two sensors are used as a function of relative humidity, and their functional expressions are expressed. The coefficient of determination R^2^ is 0.9724 and 0.9977, respectively, indicating a good curve fitting. Figure 8c,d show that the two humidity sensors have good stability in the low humidity (11.3%, 32.8%, and 57.6%) and in the high humidity (57.6%, 75.3% and 97.3%) environments, respectively.

Table 3 compares the humidity detection performance of the micro-arc MgO moisture-sensitive film layer with other sensing materials. It can be seen that various humidity sensors exhibit different humidity detection capabilities in different humidity environments. In general, the impedance is more suitable as a detection signal under low humidity conditions, and the capacitance is more suitable as a detection signal under high humidity conditions. Compared with other moisture-sensitive materials, the micro-arc MgO film layer has more obvious sensitivity difference in high-humidity and low humidity environments.

### 3.3. Humidity Sensing Mechanism Analysis

#### 3.3.1. Sensing Mechanism of Micro-Arc MgO with Impedance as the Response Signal

MgO is a p-type semiconductor, which has a larger band gap (7.8 eV) than other metal oxides. Since the surface positive ions (Mg^2+^) had greater electron affinity than the internal positive ions, the surface acceptor level occurred slightly below the conduction band bottom [35]. The negative ion (0^2−^) on the surface had greater electron repellency than the internal negative ion, so the surface donor level appeared slightly above the valence band. Before the adsorption of water molecules by micro-arc MgO, the density of surface states of donors was significantly higher than that of acceptor surface states, which led to the downward bending of the energy band at the surface, forming a hole depletion layer, and exhibiting high impedance [36]. The surface of the micro-arc MgO ceramic material was in contact with water molecules, where the electrons did not need to cross the valence band, thus the water molecules could directly source electrons from the semiconductor valence band. One end of the hydrogen atom was attracted by O^2−^ on the surface of the material, which caused the density of the surface donor state to decrease. The originally trapped holes were released, while the carrier concentration rose, showing a decrease in resistance [37]. At the same time, compared with other metal oxides, MgO had faster carrier mobility [19,35], which made the film impedance decrease rapidly with increasing humidity at the low relative humidity, showing rapid response. However, with the increase of humidity, it had been confirmed that the thickness of the water film was as high as several tens of water molecules in the case of high humidity, and when the physically adsorbed water molecules could not continue to form an additional donor (acceptor) state, that is, in a high humidity environment, the impedance of the micro-arc MgO did not continue to change significantly with the increase of the relative humidity [38].

#### 3.3.2. Sensing Mechanism of Micro-Arc MgO with Capacitance as the Response Signal

As shown in Figure 9, the capacitance of the sensor is related to the proton conductivity [19]. Firstly, H_2_O was ionized to form hydroxyl (OH^−^) and hydrogen (H^+^) [39]:(3)$$H_{2}O\Leftrightarrow H^{+}+OH^{-}$$

OH^−^ was chemically adsorbed on the surface of the membrane and was connected to the metal cation (Mg^2+^) by hydrogen bonding, so that the chemically adsorbed aqueous layer was firmly bonded to the surface of the sensing membrane, which was difficult to remove. In a low humidity environment, discontinuous water molecules were mainly physically adsorbed above the chemisorbed layer to form a first physically adsorbed aqueous layer. However, because hydrogen bonding limited the free movement of water molecules, protons did not have a continuous jump path, thus at this stage, the capacitance increased slowly with increasing RH. As RH continued to increase to higher levels, multiple continuous water molecules appeared on the surface of the membrane layer, which made the transfer of hydronium ions (H_3_O^+^) easier than before. According to the Grotthuss chain reaction [40,41], this process was under an electrostatic field where proton hopping occurred and proton conductivity was significantly improved:(4)$$2H_{2}O\to H_{3}O^{+}+OH^{-}$$
(5)$$H_{3}O^{+}+H_{2}O\to H_{2}O+H_{3}O^{+}$$

Physically adsorbed water molecules would penetrate into the pores of the membrane layer, further improving the sensor response. All of these factors caused a sudden increase in capacitance, so there was a large slope in the relationship between capacitance and RH. Meanwhile, according to this mechanism, three adsorbing steps, including the generation of first chemisorbed layer, first physisorbed layer, and multi-physisorbed layer, exist in adsorption process. Each process will continuously increase the capacitance of the sensor. Compared to a resistive humidity sensor, the response time of a micro-arc MgO humidity sensor with a capacitance as a response signal is longer.

## 4. Conclusions

For the first time, we used MAO to prepare a MgO ceramic moisture-sensitive film with a large number of micropores appearing on the surface, and we used impedance and capacitance as the response signals. The experimental results showed that in the low humidity range (11.3–67%RH) with impedance as the response signal, the average output was up to 150, with faster response (13 s) and recovery (61 s). Under the high humidity conditions (67–97.3%), with capacitance as the response signal, the average output of the micro-arc MgO humidity sensor was as high as 120. Based on the above experimental results, we can increase the humidity detection range of the sensing material by selecting an appropriate response signal for different humidity conditions, and then develop a humidity sensor with fast response and recovery, small hysteresis, and high response.

## Figures and Tables

**Figure 1 sensors-20-01736-f001:**
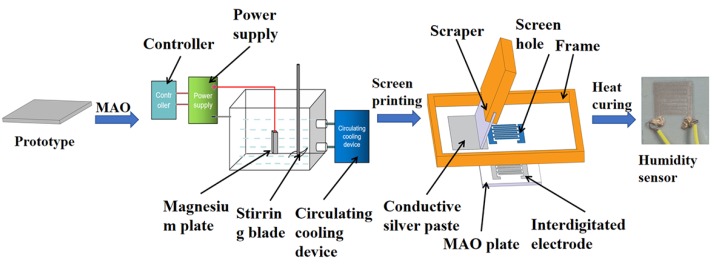
Micro-arc MgO humidity sensor fabrication process.

**Figure 2 sensors-20-01736-f002:**
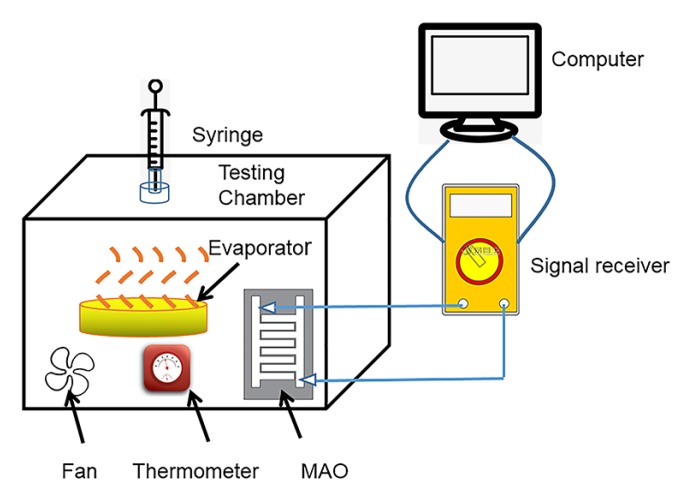
Schematic diagram of experimental setup for testing humidity sensors.

**Figure 3 sensors-20-01736-f003:**
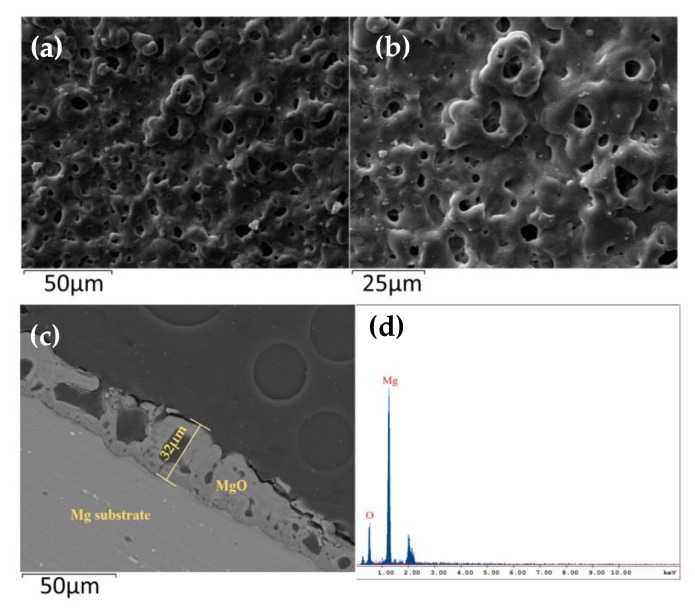
SEM images of micro-arc MgO film layer surface (**a**) low magnification and (**b**) high magnification; (**c**) section image of micro-arc MgO; (**d**) EDS image of micro-arc MgO.

**Figure 4 sensors-20-01736-f004:**
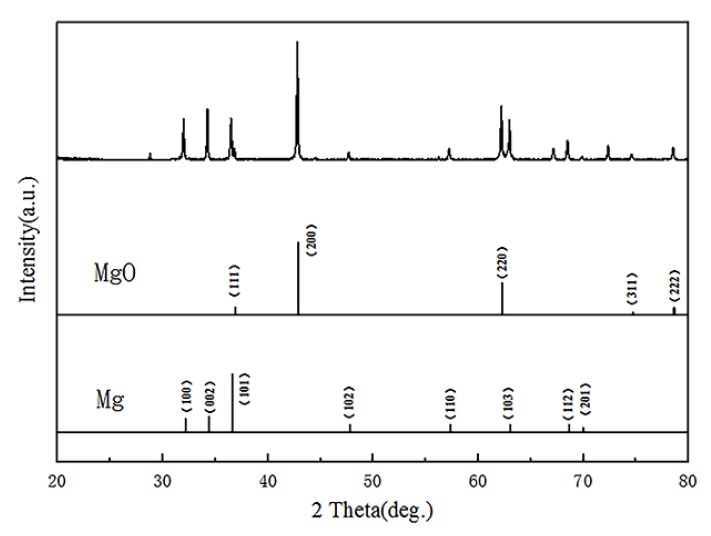
XRD image of micro-arc MgO film.

**Figure 5 sensors-20-01736-f005:**
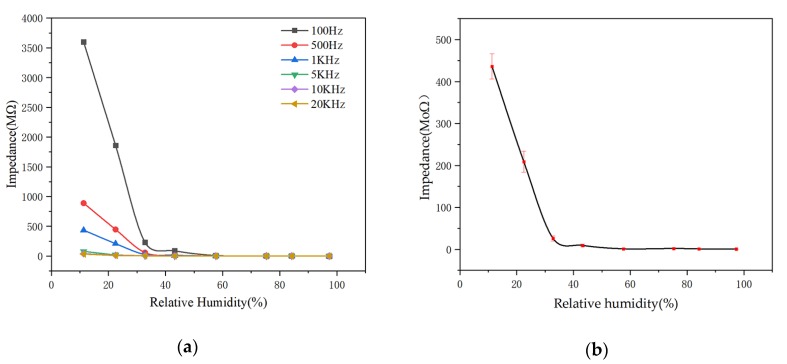
(**a**) Effect of humidity on the impedance of micro-arc MgO at different frequencies; (**b**) 1 kHz error graph.

**Figure 6 sensors-20-01736-f006:**
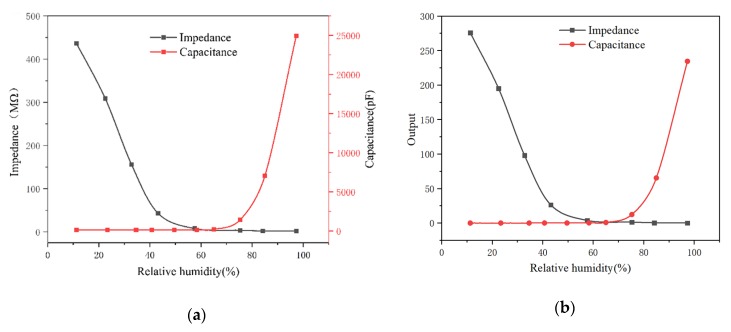
Micro-arc MgO humidity sensor (**a**) impedance and capacitance versus relative humidity; (**b**) output versus relative humidity.

**Figure 7 sensors-20-01736-f007:**
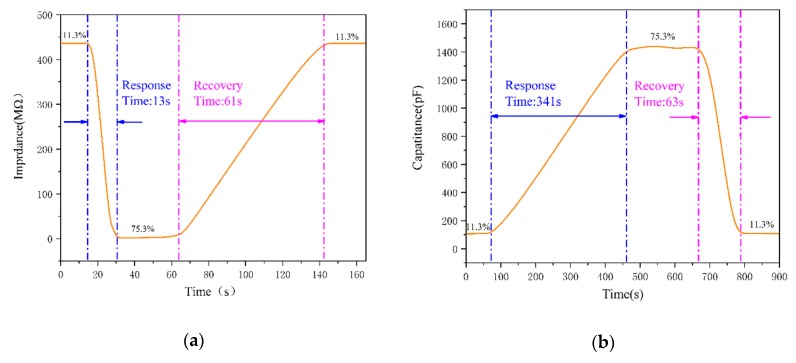
(**a**) Response/recovery time and (**c**) hysteresis characteristics of micro-arc MgO humidity sensor with impedance as response signal; (**b**) response/recovery time and (**d**) hysteresis characteristics of the micro-arc MgO humidity sensor with capacitance as a response signal.

**Figure 8 sensors-20-01736-f008:**
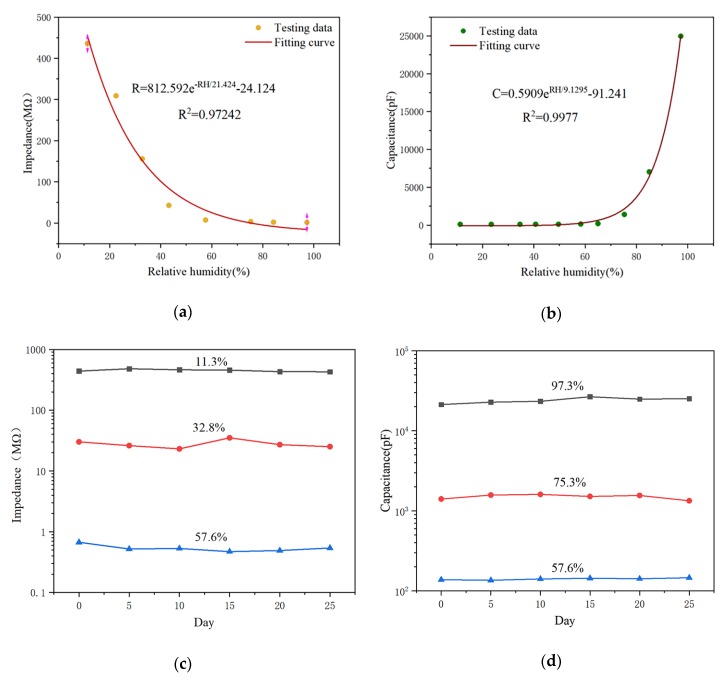
Sensor response as a function of RH in the range of 11.3–96.3% RH for the (**a**) impedance and (**b**) capacitance as micro-arc MgO humidity sensor response signal, the (**c**) impedance and (**d**) capacitance are used as the response signals of the micro-arc MgO humidity sensor, with good stability in low humidity and high humidity environments respectively.

**Figure 9 sensors-20-01736-f009:**
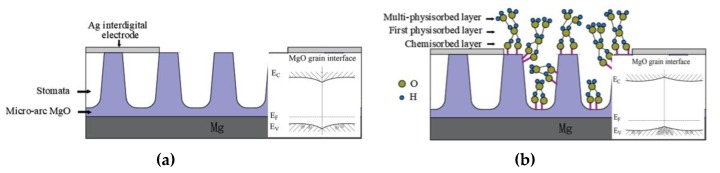
Schematic diagram of the mechanism of micro-arc MgO film surface (**a**) before water molecule adsorption and (**b**) after water molecule adsorption. The illustration in (a) is the surface energy state of the film before the adsorption of water molecules, and the illustration in (**b**) is the surface energy state of the film after the adsorption of water molecules.

**Table 1 sensors-20-01736-t001:** Response and recovery time of several moisture-sensitive materials.

Type	Meas Range	Inductive Signal Type	Response Time (s)	Recovery Time (s)	Ref
MAO MgO	11–75.3%RH	Impedance	13	61	This work
		Capacitance	341	63	
MoSe_2_/CuWO_4_	0–67%RH	Capacitance	109	9	[22]
ZnO/MoS_2_	0–85%RH	Current	138	166	[23]
PbBi_2_Se_4_	11–97%RH	Resistance	65	75	[24]
NFC/CNT	11–95%RH	Current	330	377	[25]
Polyimide	25–90%RH	Capacitance	20	22	[26]

**Table 2 sensors-20-01736-t002:** Comparison of hysteresis effects of several moisture-sensitive materials.

Type	Maximum Hysteresis (%)		Inductive Signal Type	Relative Humidity (% RH)	Ref
MAO MgO	3.8		Impedance	13	This work
	5.7		Capacitance	341	
SnO_2_/MoS_2_	5.5		Capacitance	67	[27]
TiO_2_/graphene	6.02		Resistance	48	[28]
Au-ZnO	2.35		Impedance	75	[29]
Polyaniline–holmium	1.2		Resistance	55	[30]
Gr-AgNps	6	Capacitance		80	[31]
Gr-AgNps-PMMA	9			75	

**Table 3 sensors-20-01736-t003:** Comparison in sensing properties towards various humidity sensors.

Type	Sensitivity		Inductive Signal Type	Meas. Range	Ref
	11–57% RH	57–90% RH			
MAO MgO	0.5 pF/%RH	800 pF/%RH	Capacitance	11.3–97.3% RH	This work
	9 MΩ/%RH	33 kΩ/%RH	Impedance		
AAO Al_2_O_3_	0.05 pF/%RH	0.3 pF/%RH	Capacitance	20–90% RH	[15]
	3.125 MΩ/%RH	6.67 MΩ/%RH	Resistance		
Si-NPA	3.8 kΩ/%RH	0.65 kΩ/%RH	Resistance	11.3–94.6% RH	[32]
SnO_2_/MoS_2_	333 pF/%RH	4600 pF/%RH	Capacitance	0–90% RH	[27]
AAO Al_2_O_3_	80 pF/%RH	333 pF/%RH	Capacitance	10–90% RH	[33]
CuFe_2_O_4_-Y_2_0_3_	8 MΩ/%RH	1 MΩ/%RH	Resistance	11.3–97.3% RH	[34]

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
