# Peer review of "Design and Verification of Humidity Sensors Based on Magnesium Oxide Micro-Arc Oxidation Film Layers"

_sensors, 2020, doi:10.3390/s20061736_

Round 1

Reviewer 1 Report

Mistakes exist in the submission, e.g. Figure 6(c) has no relationship with respone/recovery time. Should check and correct other obvious typos in languages and equations.

Under different humidity conditions, MgO showed different characteristics, either impedence or capacitance. Whether this phenomenon can be applied to other components need the support of more experimental results. In addition, how to design a conditional circuit to automatically switch between different working conditions?

The testing results show the response time of MgO is 13s and its recovery speed is 76s. These parameters didn't show any advantage over commercially available sensors

Author Response

The responses enclosed, Many thanks.

Reviewer 2 Report

In the manuscript “Design and verification of humidity sensor based on magnesium oxide micro-arc oxidation film layer”, the authors presented a magnesium oxide (MgO) humidity sensor based on micro-arc oxidation (MAO) technology. The sensor can simultaneously use impedance and capacitance as response signals in the range of 11.3% - 97.3%RH with fast response and recovery time.  Also, the sensor shows good stability under high and low humidity. Overall, this work is well designed, and the conclusions are well supported with sound experimental results. I suggest the paper be accepted after addressing some comments:

In the introduction, the author mentioned “… MgO as a transition metal oxide is rarely used as a sensing material, the main reason is that it has a larger band gap, which makes difficult to generate carriers. However, the faster carrier mobility makes it have the potential to develop a rapid response humidity sensor.” The author should have some discussion on the comparison of response time of this works and other reported humidity sensors. In Page 3, the author mentioned “… The film layer has good continuity, and the surface shows a large number of abnormal pores with a diameter of 4-8 μm.” During the fabrication, can the pore size be adjusted? Will the pore size affect the sensing performance? In Figure 2, the figure numbers and scale bars should be clear and readable. In Figure 3, the caption is not corresponding to the figure. In Figure 6, the figure numbers in the figure caption are not correct. Figure 6(b) should be 6(c), 6(c) should be 6(b). Please check and fix those. In Figure 6, the response and recovery times for the sensor using impedance and capacitance as response signals are very different. Is there any explanation for the difference? In Figure 6(a), the impedance of the sensor at 11.3%RH is 400-500MOhm. However, in Figure 7(c), the impedance of the sensor at 11.3%RH is above 1000MOhm. Why are they so different?

Author Response

Please find the response attached. Thanks.

Reviewer 3 Report

The authors present a humidity sensor based on a MgO sensing film which was fabricated using the micro-arc oxidation technique. The designed sensor uses both sensor impedance and capacitance as a response signal for humidity detection based on the operation range. However, the article requires major revisions which should be done before the article is considered for publication. Some of my suggestions to improve the quality of the manuscript are –

Suggest revision to the references. There are multiple occurrences where the provided references are not relevant or where references are missing. It is difficult to understand the fabrication process and the test setup of the sensor. Suggest adding more details about the fabrication procedure and also a brief introduction about MAO to help non-specialist readers. What is the thickness of the formed MgO layer? In Figure 4, Please add error bars to show noise levels during measurements. A comment regarding the repeatability of the results would also be helpful. At higher humidity levels, the impedance is nearly zero at all frequencies. Can the authors comment on this? Equations (1) and (2) are not sensitivities but are the normalized change in sensor output. Revise Fig. 5 and the text accordingly. In Fig. 6b, The %RH starts at 11% and is increased to 75%RH. The sensor response here is linear and does not agree with the data shown in Fig 6b where the sensor is almost non-responsive in the low humidity range. A control signal plot should be added to both Fig. 6a and 6b to show how the humidity of the test chamber changes with time. The authors should clarify why the response time of the sensor is different (13s and 341s) when using impedance and capacitance. The choice of response signal should have no effect on the response time of the device. The writing quality of the paper needs to be significantly improved.

Author Response

(The authors gave the same response as above.)

Round 2

Reviewer 1 Report

Using material's capacitance/resistance to develop humidity sensors is a common means in practical applications, however, the key point is to improve the product performance (high accuracy, quick response, low hysteresis,etc). So I would suggest to compare different materials' charactrisitcs in terms of the above-mentioned characteristics. In addition, a practical usage (even a simple prototype circuit) is necessary to prove the efficacy.

More experiments are needed, especially trials that can prove the performance of the product instead of a pure research on material's characteristics

Author Response

The responses is attached, thanks.

Reviewer 3 Report

Dear authors,

Thank you for submitting the revised article and addressing the reviewer's comments.

I would recommend the article for publishing after a few minor revisions. 

1) Minor language corrections and spell checks required throughout the manuscript.

2) It would be helpful if the authors can improve the quality (resolution) of the figures and graphs in the manuscripts to improve readability.

Author Response

The responses is attached, thanks.

Round 3

Reviewer 1 Report

If the developed sensor type was compared with commercially available products, it would be better.
